# The Impact of COVID-19 Era on Pulmonary Embolism Patients: Increased Incidence of Hospitalizations and Higher Mortality—What Can Be Done?

**Aura Vîjîiac** [1,2], **Diana Irena Stănciulescu** [2,*], **Alexandru Emil Băetu** [1,2], **Iulia-Adelina Grigore** [3], **Denisa Vintilă** [2], **Cosmin Cojocaru** [1,2], **Elisabeta Bădilă** [1,2], **Horatiu Moldovan** [1,2] and **Alexandru Scafa-Udriște** [1,2]

1   Carol Davila University of Medicine and Pharmacy, 050474 Bucharest, Romania;
    aura.vijiiac@drd.umfcd.ro (A.V.); alexandru.baetu@rez.umfcd.ro (A.E.B.);
    cojocaru.cosmin@drd.umfcd.ro (C.C.); elisabeta.badila@umfcd.ro (E.B.); h_moldovan@hotmail.com (H.M.);
    alexandru.scafa@umfcd.ro (A.S.-U.)
2   Emergency Clinical Hospital of Bucharest, 014461 Bucharest, Romania; elena-denisa.vintila@rez.umfcd.ro
3   St. Pantelimon Emergency Hospital of Bucharest, 014461 Bucharest, Romania;
    iulia-adelina.grigore@rez.umfcd.ro
*   Correspondence: diana-irena.stanciulescu@rez.umfcd.ro; Tel.: +40-21-599-2264

**Abstract:** The coronavirus disease outbreak in 2019 (COVID-19) reached devastating pandemic proportions, still representing a challenge for all healthcare workers. Furthermore, the social environment underwent significant changes and healthcare facilities were overwhelmed by COVID-19 patients. The purpose of our study was to compare the prevalence, characteristics and outcomes of 234 patients presenting with pulmonary embolism diagnosed by computed tomography pulmonary angiography (CTPA) during the COVID-19 pandemic, to patients presenting with PE one year before, aiming to assess differences and similarities between these patients. Or main findings were: patients with PE had worse survival during the pandemic, there was an increased incidence of PE among hospitalizations in our cardiology unit during the COVID-19 pandemic, and patients hospitalized with PE during the pandemic were more likely to be obese, allowing us to infer that weight control can have a positive impact on preventing PE. Future research should establish optimal therapeutic, epidemiological and economical strategies for non-COVID patients, as the pandemic continues to put significant burden on the healthcare systems worldwide.

**Keywords:** pulmonary embolism; COVID-19 pandemic; venous thromboembolism

## 1. Introduction

Pulmonary embolism (PE) is a potentially fatal disease, with heterogenous clinical presentation and outcome. A wide variety of predisposing genetic and environmental factors interact in the pathophysiological continuum of venous thromboembolism [1]. Identifying each patient's risk factors (either irreversible or transient) is essential for assessing the risk of recurrence and for adopting a patient-tailored antithrombotic strategy.

The coronavirus disease outbreak in 2019 (COVID-19) reached devastating pandemic proportions, spreading all over the globe and still representing a challenge for all healthcare workers. Although knowledge on the COVID-19 infection is growing fast, therapy is still mostly empirical, and the complications of the disease are still not completely understood [2–4]. Due to the pandemic outbreak, the social environment also underwent significant changes. Lockdowns and overwhelmed healthcare facilities hindered patients' access to medical assistance. Since the pandemic outbreak, an increasing number of studies have shown abnormal coagulation parameters in patients hospitalized with severe forms of COVID-19 [5–7]. Furthermore, critically ill patients undergo prolonged immobilization and therefore the risk of thromboembolic events is higher [8]. So far, the characteristics of

patients with PE during the COVID-19 pandemic and the prevalence of PE in COVID-19 patients hasscarcely been investigated.

The purpose of our study was to compare the prevalence, characteristics and outcomes of patients presenting with PE during the COVID-19 pandemic to patients presenting with PE one year before.

## 2. Materials and Methods

### 2.1. Study Population

We retrospectively enrolled 234 consecutive patients with PE (either as the main or asecondary diagnostic), who were admitted to the cardiology unit of a tertiary center between January 2019 and December 2020. Our center is an emergency hospital which serves the population of Bucharest and the south-east of Romania [9]. PE was diagnosed using computed tomography pulmonary angiography (CTPA) and defined as one or more filling defects down to the level of segmental arteries [1]. The study cohort was divided in two sections: a "pre-pandemic" group, including patients with PE admitted between January 2019 and December 2019, and "pandemic" group, including patients with PE admitted between January 2020 and December 2020. The rationale for this division was an aim to assess differences and similarities between patients with PE before and during the COVID-19 pandemic. The total number of patients admitted in the cardiology unit for each month of 2019 and 2020 was also collected from hospital registries.

### 2.2. Data Collection

Basic demographic and clinical data were collected from consecutive patients admitted to our hospital. Symptoms on admission such as chest pain, dyspnea, or syncope, were collected from the patients' medical charts. Shock on admission was defined as hemodynamic instability (namely, systolic blood pressure [BP] <90 mm Hg or vasopressors required to maintain BP >90 mm Hg) with signs of end-organ hypoperfusion (e.g., altered mental status, oliguria, cold skin) [1]. The Pulmonary Embolism Severity Index (PESI) score was calculated on admission according to current guidelines. All patients were first admitted to the cardiology acute intensive care and monitoring unit, regardless of their hemodynamic status.

We recorded the presence of predisposing factors for PE such as obesity, prolonged immobility, previous venous thromboembolism, history of atrial fibrillation, thrombophilia (previously known by the patient), or cancer (either previously known or diagnosed throughout the current hospitalization). All patients who had no obvious cause of PE were referred at discharge to a hematology center for thrombophilia testing.

For patients admitted during the COVID-19 pandemic, their COVID-positive or negative status was assessed using real time quantitative reverse-transcription polymerase chain reaction (RT-PCR) assay on throat/nose swabs, which were routinely performed for all patients admitted in the cardiology unit, according to the hospital's epidemiologic strategy. Previous history of SARS-CoV-2 infection was also assessed.

Standard 12-lead electrocardiograms were performed on admission for all patients.

We assessed the signs of right ventricular impairment on the electrocardiogram (ST-segment or T-wave changes in the precordial leads, $S_1Q_3T_3$ pattern) and the presence of newly developed right bundle branch block.

All patients underwent standard transthoracic echocardiography (TTE) acquisitions. Left ventricular ejection fraction (LVEF) was measured using the Simpson biplane method in the apical 4-chamber and 2-chamber views [10].

The right ventricular (RV) end-diastolic diameter and the right atrial (RA) end-systolic diameter were assessed from the apical 4-chamber view. The gradient between the RV and the RA was obtained from the spectral Doppler of the tricuspid regurgitation (TR) jet. The tricuspid annular plane systolic excursion (TAPSE) was measured with M-mode in the apical 4-chamber view as the systolic longitudinal displacement of the tricuspid annular plane [1]. The presence of intracardiac thrombi was also assessed with TTE.

Data obtained from CTPA were used to record the extent and localization of the filling defects in the pulmonary arterial bed, as well as the co-existence of pulmonary infarction and pneumonia lesions (the latter, particularly in patients with confirmed SARS-CoV-2 infection). Patients with clinical signs of deep venous thrombosis (DVT) underwent venous lower extremity Doppler ultrasound and if DVT was confirmed, the extent and localization of peripheral venous thrombi were also recorded.

We collected data regarding the anti-thrombotic treatment received-fibrinolytics when indicated [1], anticoagulation throughout hospitalization, anticoagulant regimen at discharge. The duration of hospitalization was also recorded for each patient. The study protocol complied with the Declaration of Helsinki and was approved by the human research committee of our hospital.

### 2.3. Statistical Analysis

Statistical analysis was performed using the SPSS version 20.0 statistical software package. The normality of variables was evaluated using the Kolmogorov–Smirnov test. Continuous data are presented as mean ± standard deviation if normally distributed and compared using the Student's t-test. Continuous data are displayed as median and interquartile range if not normally distributed and compared using the Mann–Whitney Utest. Categorical data are presented as numbers and percentages, and compared using the Fisher exact test, depending on normality. *p*-values < 0.05 were considered statistically significant.

Multivariate Cox regression analysis was used to evaluate the role of the clinical presentation and of predisposing factors. Survival time was used for this analysis, and the endpoint was represented by the occurrence of in-hospital death. By using Kaplan–Meier analysis, survival curves during the pre-pandemic and the pandemic period were compared with the log-rank test.

### 3. Results

Between January 2019 and December 2019, 4595 patients were admitted to the cardiology unit, while between January 2020 and December 2020, only 2404 patients were hospitalized in the cardiology department (representing 52.3% of the total admissions in 2019). Among those, 107 (2.3%) patients from 2019 and 127 (5.3%) patients from 2020 had PE, respectively (*p* < 0.0001), reflecting a higher incidence of PE among hospitalizations in the cardiology unit in the year when the COVID-19 pandemic was declared (Figure 1). The final study cohort thus included 234 consecutive patients with PE, divided into "pre-pandemic" (107 patients) and "pandemic" groups (127 patients). Baseline demographic and clinical characteristics are summarized in Table 1.

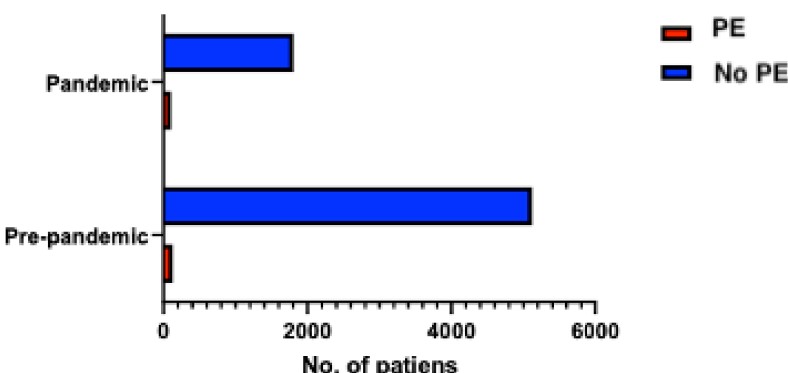

**Figure 1.** Distribution of patients with PE among the total number of hospitalized patients during the pre-pandemic (107 patients with PE) and pandemic (127 patients with PE) periods.

**Table 1.** Baseline characteristics of the two groups.

| Variables | Pre-Pandemic Group (*n* = 107) | Pandemic Group (*n* = 127) | *p*-Value |
|---|---|---|---|
| Age (years) | 61 ± 16 | 61 ± 17 | 0.77 |
| Men, *n* (%) | 71 (66%) | 61 (48%) | **0.007** |
| Symptoms on Admission | | | |
| Syncope, *n* (%) | 8 (7%) | 17 (13%) | 0.14 |
| Dyspnea, *n* (%) | 75 (70%) | 86 (68%) | 0.70 |
| Chest pain, *n* (%) | 33 (31%) | 31 (24%) | 0.27 |
| Hemodynamic instability, *n* (%) | 1 (1%) | 2 (2%) | 0.66 |
| PESI score | 88.8 ± 30.5 | 97.0 ± 43.2 | 0.09 |
| Predisposing Factors | | | |
| Obesity, *n* (%) | 16 (15%) | 54 (43%) | **<0.001** |
| Prolonged immobility, *n* (%) | 16 (15%) | 15 (12%) | 0.48 |
| Atrial fibrillation, *n* (%) | 12 (11%) | 13 (10%) | 0.81 |
| Cancer, *n* (%) | 26 (24%) | 24 (19%) | 0.32 |
| Thrombophilia, *n* (%) | 9 (8%) | 9 (7%) | 0.70 |
| History of previous PE, *n* (%) | 7 (7%) | 8 (6%) | 0.85 |
| Deep vein thrombosis, *n* (%) | 51 (48%) | 66 (52%) | 0.51 |
| Days of hospitalization | 10 (7–12.25) | 8 (6–11) | 0.07 |

Continuous data are expressed as mean ± standard deviation. Categorical data are expressed as number (percentage). *n*—number of patients; PE—pulmonary embolism; PESI—pulmonary embolism severity index. Bolded *p*-values are statistically significant.

Regarding the location of filling defects, there was no statistically significant difference between the pre-pandemic and pandemic group (*p* = 0.25). Computed Tomography Angiography (CTPA) results are plotted in Figure 2.

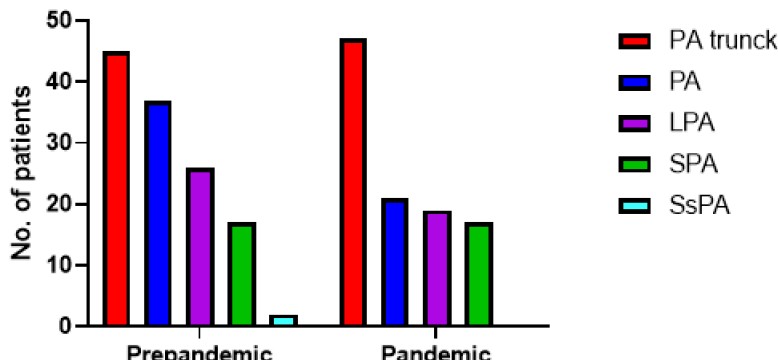

**Figure 2.** Distribution of filling defects of the patients with PE during the pre-pandemic (107 patients with PE) and pandemic (127 patients with PE) periods. PA trunk = pulmonary artery trunk (*n* = 45 vs. *n* = 47), PA = unilateral pulmonary artery (*n* = 37 vs. *n* = 21), LPA = lobular pulmonary artery (*n* = 26 vs. *n* = 19), SPA = Segmental pulmonary artery (*n* = 17 vs. *n* = 17), SsPA = Subsegmental pulmonary artery (*n* = 2 vs. *n* = 0).

PE occurred mostly in men during the pre-pandemic period (66% men, 34% women), but the distribution of PE among sexes during the pandemic period was similar (48% men, 52% women, *p* = 0.007) (Figure 3). There were no differences in terms of PE severity between the pre-pandemic and pandemic group (*p* = 0.66 for hemodynamic instability, *p* = 0.09 for PESI score). From the pandemic group, 9 patients (7%) tested positive for SARS-CoV-2 and 6 patients (5%) had a positive history of COVID-19 infection.

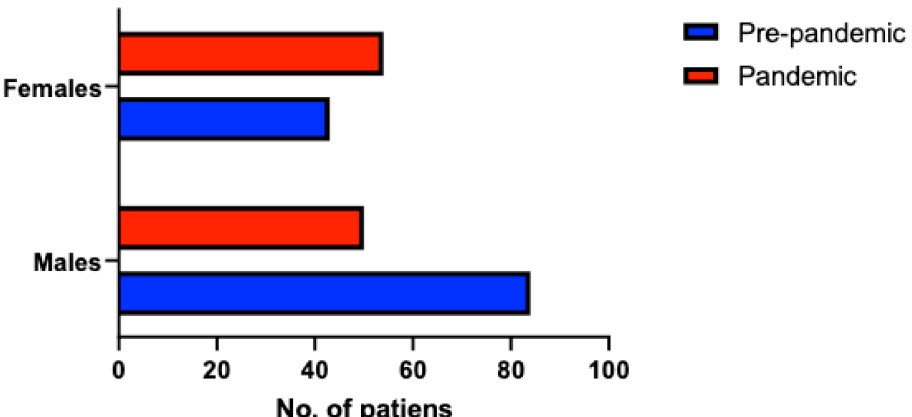

**Figure 3.** Distribution of patients with PE according to sexes during the pre-pandemic and pandemic period.

We performed multivariate Cox regression for clinical presentation. The analysis has an overall model fit described by a Null-Model-2 Log Likelihood of 142.835, and a Full-Model-2 Likelihood of 134,073 with a Chi-Squared value of 8.762 ($p$ = 0.03). The analysis of beta coefficients that defines the degree and direction of change in the outcome did not reveal any symptoms of presentation as a determinant of the outcome (dyspnea-beta coefficient = 0.23, $p$ = 0.72, chest pain-beta coefficient = −13.83, $p$ = 0.96, syncope-beta coefficient = −0.19, $p$ = 0.85), therefore the clinical presentation was similar between the two groups.

The second multivariate Cox regression was performed for predisposing factors. This analysis has an overall model fit described by a Null-Model-2 Log Likelihood of 143,105, a Full-Model-2 Likelihood of 122,534 with a Chi-Squared value of 20.573 ($p$ = 0.008). The analysis of beta coefficients that defines the degree and direction of change in the outcome does not reveal any predisposing factors as a determinant of the outcome (history of Covid-beta coefficient t = −9.79, $p$ = 0.96, Covid positive-beta coefficient = 1.23, $p$ = 0.16, atrial fibrillation-beta coefficient = −0.10, $p$ = 0.87, prolonged immobility-beta coefficient = 0.71, $p$ = 0.34, cancer-beta coefficient = 0.13, $p$ = 0.83, obesity-beta coefficient = 0.23, $p$ = 0.71, non-Covid pneumonia beta coefficient = 0.16, $p$ = 0.78).

Obesity was a more frequent predisposing factor among the pandemic group, although it is not significant as a determinant of the outcome of patients with PE.

The prevalence of co-existent deep vein thrombosis did not differ significantly between the pre-pandemic and the pandemic group ($p$ = 0.51).

There were no significant differences between the two groups in terms of ECG (Table 2) or echocardiographic changes (Table 3).

**Table 2.** ECG changes in the two groups.

| ECG Changes | Pre-Pandemic Group ($n$ = 107) | Pandemic Group ($n$ = 127) | $p$-Value |
|---|---|---|---|
| Newly developed RBBB, $n$ (%) | 17 (16%) | 19 (15%) | 0.84 |
| ST elevation in precordial leads, $n$ (%) | 0 (0%) | 1 (1%) | 0.55 |
| T wave inversion in precordial leads, $n$ (%) | 20 (19%) | 38 (30%) | 0.07 |
| S1Q3T3 pattern, $n$ (%) | 22 (21%) | 19 (15%) | 0.34 |

Categorical data are expressed as number (percentage). $n$—number of patients; ECG—electrocardiogram; RBBB—right bundle branch block; S1Q3T3—S wave in lead I, Q wave in lead III, inverted T wave in lead III.

**Table 3.** Echocardiographic changes in the two groups.

| Echocardiographic Changes | Pre-Pandemic Group (n = 107) | Pandemic Group (n = 127) | p-Value |
|---|---|---|---|
| Intracardiac thrombus, *n* (%) | 6 (6%) | 7 (6%) | 0.80 |
| TAPSE (mm) | 20 ± 4 | 20 ± 4 | 0.74 |
| RV diameter (mm) | 37 ± 10 | 38 ± 10 | 0.25 |
| RA diameter (mm) | 37 ± 9 | 37 ± 9 | 0.74 |
| RV-RA gradient (mm Hg) | 30 ± 12 | 30 ± 16 | 0.96 |

Continuous data are expressed as mean ± standard deviation. Categorical data are expressed as number (percentage). *n*—number of patients; TAPSE—tricuspid annular plane systolic excursion; RV—right ventricle; RA—right atrium.

There were four in-hospital deaths in the pre-pandemic group (4%) and 14 in-hospital deaths in the pandemic group (11%), two of the latter occurring in patients who tested positive for SARS-CoV-2 on admission. None of the recorded in-hospital deaths occurred in patients with a previous history of COVID-19. In the Kaplan–Meier analysis, patients in the pandemic group had worse in-hospital survival than patients in the pre-pandemic group (log-rank test, *p* = 0.04) (Figure 4).

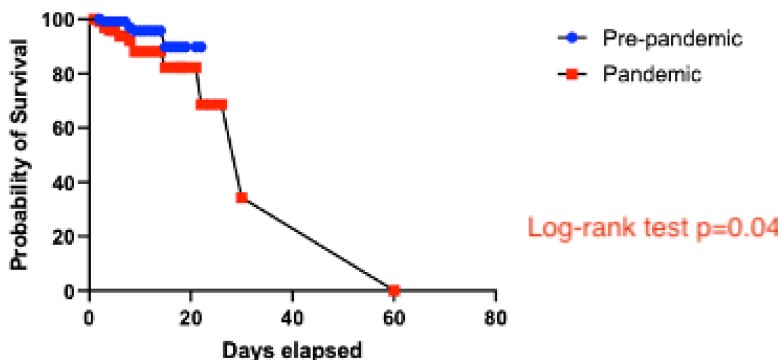

**Figure 4.** Survival analysis comparing the pre-pandemic and the pandemic groups.

A comparison of the anticoagulant treatment recommended for patients who were discharged alive is shown in Table 4.

**Table 4.** Antithrombotic regimen at discharge in the two groups.

| Antithrombotic Treatment at Discharge | Pre-Pandemic Group (n = 103) | Pandemic Group (n = 113) | p-Value |
|---|---|---|---|
| Anti-vitamin K, *n* (%) | 30 (28%) | 32 (25%) | 0.98 |
| Rivaroxaban, *n* (%) | 32 (30%) | 29 (23%) | 0.47 |
| Apixaban, *n* (%) | 29 (27%) | 38 (30%) | 0.47 |
| Dabigatran, *n* (%) | 5 (5%) | 1 (1%) | 0.17 |
| LMWH, *n* (%) | 7 (7%) | 13 (10%) | 0.34 |

Categorical data are expressed as number (percentage). *n*—number of patients; LMWH—low molecular weight heparin.

## 4. Discussion

The main findings of our study are:

1. Patients with PE had worse survival during the pandemic.
2. There was an increased incidence of PE among hospitalizations in our cardiology unit during the COVID-19 pandemic;
3. Patients hospitalized with PE during the pandemic were more likely to be obese;

SARS-CoV-2 infection induces a form of intravascular coagulopathy which predisposes to thrombosis [7], but whether these hemostatic changes are a direct effect of the virus or derive from the cytokine activation and release is incompletely understood. Consequently, patients with both active or previous SARS-CoV-2 infection are at risk for thrombotic and thromboembolic events, as the systemic inflammation and pro-coagulant status are believed to persist long after the resolution of the viral infection [11]. Similar to previous reports [12,13], we found a significant increase in the incidence of PE among hospitalizations in our cardiology unit during the COVID-19 pandemic. However, only 12% of our pandemic group had either active or previous SARS-CoV-2 infection, reflecting a higher risk for PE for the non-COVID-19 population as well. Due to cancellation of elective medical procedures, strict social shutdown and fear of the virus, the number of patients seeking medical attention during the pandemic decreased worldwide [14–17]. It is widely believed that the lockdown has caused collateral damage in non-COVID-19 patients [12,18], who either sought medical care very late or could not receive proper timely intervention because most of the resources had been re-directed to fighting the pandemic [19]. Moreover, prolonged social restrictions during the COVID-19 pandemic led to decreased physical activity, which in turn had a deleterious effect on the obesity epidemic [20,21]. Obesity is an established predisposing factor for PE [1] and it was a significant determinant of PE during the pandemic in our study, however it had no influence on the outcome.

In our cohort, PE was significantly more lethal during the pandemic than during the year before [9]. This happened despite the fact that only a small minority of the recorded deaths occurred in COVID-19 patients. A probable explanation for this increased mortality is the patients' hesitation to seek medical help during the pandemic, which leads to delayed diagnosis and worsening of the disease. Early diagnostic is crucial in PE, since it allows prompt initiation of antithrombotic treatment. Another possible explanation could be the overwhelmed emergency departments and intensive care units during the time of pandemic, which might have hindered some patients from receiving adequate and timely procedures and therapies [19,22]. Since PE and COVID-19 share some overlapping symptoms such as chest pain and dyspnea, it can be argued that the diagnostic of PE might be delayed if the patient is suspected as having COVID-19 and investigated accordingly [18]. Although all patients were tested for SARS-CoV-2 on admission, we cannot rule out false negative cases as a potential contributor to increased mortality.

Watchmaker et al., published a retrospective review of PE patients from six emergency hospitals from New York City in which they compared non-pandemic and pandemic PE admissions from 1 March–1 April 2019 and 1 March 1–1April 2020, and showed a 10.9% increase of PE presentations in the pandemic era and no enrichment in the traditional risk factors [13].

V. Tilliridou et al., published a retrospective audit in July 2021, which assessed the frequency and severity of pulmonary embolism in 2020 compared to 2019: patients with both PE and COVID-19 had an increased 30-day mortality and worse outcomes compared to those without COVID-19 (28% versus 6%) [23]. Moreover, delays due to COVID-19 protocols all over the world were reported to forestall the treatment for acute pathologies [24].

In a retrospective multi-sourced nationwide cohort study of adults admitted to hospital with thromboembolic events (TE) and deaths from TE in England between 1 February 2018 and 31 July 2020, Aktaa et al., found an increased incidence of TE, particularly PE, despite only a small proportion of the cohort testing positive for COVID-19 and also an increased mortality. This may highlight the adverse social effects of the pandemic and also the delay in seeking medical care which may lead to worse outcomes in the patients with thromboembolic events [25].

## 5. Study Limitations

The main limitation of our study comes from its retrospective, single-center design. There is also a risk of selection bias since lockdown and fear of nosocomial infections might have driven some patients with PE to not contact the health system, even if they were

having symptoms. However, due to its implications, our study should encourage further research on the subject.

## 6. Conclusions

In our cohort, there was an increased incidence of PE hospitalizations and high in-hospital mortality for these patients during the COVID-19 pandemic. Among them, obesity was statistically significant as a risk factor for the development of pulmonary embolism in the pandemic period, but without influence on mortality. Therefore, we can infer that weight control can have a positive impact.

Future research should establish optimal therapeutic, epidemiological and economical strategies for non-COVID patients, as the pandemic continues to put significant burden on the health systems worldwide.

**Author Contributions:** Conceptualization, A.S.-U., E.B., H.M.; methodology, A.S.-U.; software, A.E.B., C.C.; validation, A.S.-U. and E.B.; formal analysis, A.E.B., C.C.; investigation, A.S.-U.; resources, A.S.-U.; data curation, I.-A.G., D.V.; writing—original draft preparation, A.V., D.I.S.; writing—review and editing, A.V., D.I.S.; visualization, D.I.S.; supervision, A.S.-U.; project administration, A.S.-U.; funding acquisition, A.S.-U. All authors have read and agreed to the published version of the manuscript.

**Funding:** This research received no external funding.

**Institutional Review Board Statement:** The study was conducted according to the guidelines of the Declaration of Helsinki, and approved by the Ethics Committee of Emergency Clinical Hospital of Bucharest (request number 26198/July 2021).

**Informed Consent Statement:** Informed consent was obtained from all subjects involved in the study.

**Conflicts of Interest:** The authors declare no conflict of interest.

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
