# Peer review of "The Impact of COVID-19 Era on Pulmonary Embolism Patients: Increased Incidence of Hospitalizations and Higher Mortality—What Can Be Done?"

_covid, doi:10.3390/covid1010030_

Round 1

Reviewer 1 Report

The manuscript is of interest and timely topic. However, I have some major concerns regarding the statistical methods, results and variables considered. Therefore, the manuscript require an intense major revision. Here some suggestions:

Methods: Instead of MSCT, I suggest to use the term: Computed Tomography Pulmonary Angiography (CTPA)

Statistic: “Logistic regression analysis was used to evaluate the role of the clinical presentation and of predisposing factors to predict PE during the pandemic period. Results are re-ported as odds ratios (OR) with 95% confidence intervals (CI). By using Kaplan-Meier analysis, survival curves during the pre-pandemic and the pandemic period were com- pared with the log-rank test.” If you have a time variable, perform the multivariate regression using a Multivariate Cox Regression analysis, otherwise it has no sense to express the data with OR (timely-independent) and K-M curve with log-rank (Timely-dependent).

Among the risk factors for acute PE, it would be important to include also surgery within 4 weeks and trauma events.

Please define the criteria used for thrombophilia. Did you also assess the presence of thrombophilic conditions also after the diagnosis of PE (new case of Thrombophilia)?

Regarding the ECG characteristics, how many patients have tachycardia? How many patients have right ventricular strain (defined as the presence of at least one of the following: RBBB, S1Q3T3 and Negative T waves from V1 to V4? Data regarding the P pulmonary wave? qR in V1? Right axial deviation? All these ECG signs are associated with worst outcome.

Very few patients (just one) had a high-risk PE in the pre-pandemic era and it sound strange, being these patients about 15% of acute PE. Please discuss.

Data regarding the presence of right ventricular dysfunction at transthoracic echocardiography are needed (at least the number of patients have in a RV/LV ratio > 0.9 in 4 chamber apical view).

The Kaplan Maier curve is at the limit of statistical significance. I suggest to split the population according to the risk category of acute PE according to the ESC classification.

It would be important to describe the location of filling defects observed at CT and compare these findings between the two groups.

How many patients required ICU admission? Only two during the pandemic? Please carefully check the data

How many patients admitted to general ward and ICU developed acute PE?

The discussion must be revised considering the new findings obtain after revising the manuscript, also focusing on the difference between the risk factor for acute PE as well as the prognostic significance of Echo signs (some recent meta-analysis have been just publicised on this issue).

The manuscript must be entirely revised for English

Author Response

Response to reviewer no. 1

Thank you for your review. We have updated the manuscript and also provided answers to your observations, we hope this information finds you well and suits your concerns.

  1. “Methods: Instead of MSCT, I suggest to use the term: Computed Tomography Pulmonary Angiography (CTPA)”

Response: The terms have been modified;

  1. “Statistic: “Logistic regression analysis was used to evaluate the role of the clinical presentation and of predisposing factors to predict PE during the pandemic period. Results are re-ported as odds ratios (OR) with 95% confidence intervals (CI). By using Kaplan-Meier analysis, survival curves during the pre-pandemic and the pandemic period were com- pared with the log-rank test.” If you have a time variable, perform the multivariate regression using a Multivariate Cox Regression analysis, otherwise it has no sense to express the data with OR (timely-independent) and K-M curve with log-rank (Timely-dependent).”

Response – rows 153-170: We performed multivariate Cox Regression for clinical presentation. The analysis has an overall model fit described by a Null-Model-2 Log Likelihood of 142.835, a Full-Model-2 Likelihood of 134,073 with a Chi-Squared value of 8.762 (p=0.03). The analysis of beta coefficients that defines the degree and direction of change in the outcome does not reveal any symptoms of presentation as a determinant of the outcome (dyspnea-beta coefficient=0.23, p=0.72, chest pain-beta coefficient= -13.83, p=0.96, syncope-beta coefficient= -0.19, p=0.85), therefore the clinical presentation was similar between the two groups.

The second Multivariate Cox Regression was performed for predisposing factors. This analysis has an overall model fit described by a Null-Model-2 Log Likelihood of 143,105, a Full-Model-2 Likelihood of 122,534 with a Chi-Squared value of 20.573 (p=0.008). The analysis of beta coefficients that defines the degree and direction of change in the outcome does not reveal any predisposing factors as a determinant of the outcome (history of Covid- beta coefficient= -9.79, p=0.96, Covid positive- beta coefficient=1.23, p=0.16, atrial fibrillation- beta coefficient= -0.10, p=0.87, prolonged immobility- beta coefficient=0.71, p=0.34, cancer- beta coefficient= 0.13, p=0.83, obesity- beta coefficient= 0.23, p=0.71, non-Covid pneumonia beta coefficient=0.16, p=0.78).

Obesity was a more frequent predisposing factor among the pandemic group, although it is not significant as a determinant of the outcome of patients with PE.

  1. Among the risk factors for acute PE, it would be important to include also surgery within 4 weeks and trauma events.

Response – rows 71-72: Our center is an emergency hospital and all pulmonary embolism admissions were through the emergency department; no patient was diagnosed with PE while already admitted to general ward/other departments. We considered that prolonged immobilization includes also surgery and trauma prior to admission.

  1. Please define the criteria used for thrombophilia. Did you also assess the presence of thrombophilia conditions also after the diagnosis of PE (new case of Thrombophilia)?

Response – rows 72-76: “We recorded the presence of predisposing factors for PE such as obesity, prolonged immobility, previous venous thromboembolism, history of atrial fibrillation, thrombophilia (previously known by the patient), or cancer (either previously known or diagnosed throughout the current hospitalization). All the patients who had no obvious cause of PE were referred at discharge to a hematology center for thrombophilia testing.”

We do not have a hematology department in our hospital or the any thrombophilia testing available, which is why we only considered previously known thrombophilia;

  1. Regarding the ECG characteristics, how many patients have tachycardia? How many patients have right ventricular strain (defined as the presence of at least one of the following: RBBB, S1Q3T3 and Negative T waves from V1 to V4? Data regarding the P pulmonary wave? qR in V1? Right axial deviation? All these ECG signs are associated with worst outcome.

Response: there was no data collected regarding heart rate, only the presence of right ventricular impairment – rows 81-84: “We assessed the signs of right ventricular impairment on the electrocardiogram (ST-segment or T-wave changes in the precordial leads, S1Q3T3 pattern) and the presence of newly developed right bundle branch block.”

Rows 404-405: There were no significant differences between the two groups in terms of ECG (Table 2) or echocardiographic changes (Table 3).

  1. Very few patients (just one) had a high-risk PE in the pre-pandemic era and it sound strange, being these patients about 15% of acute PE. Please discuss.

We defined hemodynamic instability at presentation as systolic blood pressure < 90 mmHg with signs of end-organ hypoperfusion, therefore only one patient in the pre-pandemic group met these criteria (1/107) while only two patients in the pandemic group had hemodynamic instability at admission (2/127);

  1. Data regarding the presence of right ventricular dysfunction at transthoracic echocardiography are needed (at least the number of patients have in a RV/LV ratio > 0.9 in 4 chambers apical view).

We had no significant differences between the two groups;

  1. The Kaplan Maier curve is at the limit of statistical significance. I suggest to split the population according to the risk category of acute PE according to the ESC classification.

Response – row 143-168;

  1. It would be important to describe the location of filling defects observed at CT and compare these findings between the two groups.

Response – row 133-141: results have been updated; “Regarding of the location of filling defects, there was no statistically significant difference between the pre-pandemic and pandemic group (p=0.25). Computed Tomography Angiography (CTPA) results are plotted in figure 2.”

Figure 2 shows the distribution of filling defects of the patients with PE during the pre-pandemic (107 patients with PE) and pandemic (127 patients with PE) periods. PA trunk=pulmonary artery trunk (n=45 vs. n=47), PA=unilateral pulmonary artery (n=37 vs. n=21), LPA=lobular pulmonary artery (n=26 vs n=19), SPA=Segmental pulmonary artery (n=17 vs n=17), SsPA=Subsegmental pulmonary artery (n=2 vs n=0).

  1. How many patients required ICU admission? Only two during the pandemic? Please carefully check the data

Response – row 68-70: All patients were first admitted to the cardiology acute intensive care and monitoring unit according to the protocol in our department, regardless of their hemodynamic status. No patient was admitted to the general intensive care unit.

  1. How many patients admitted to general ward and ICU developed acute PE?

All the patients with pulmonary embolism presented or were brought by emergency personnel to our Emergency Department and admitted to cardiology; nobody developed acute PE while already admitted in other departments; 

  1. The discussion must be revised considering the new findings obtain after revising the manuscript, also focusing on the difference between the risk factor for acute PE as well as the prognostic significance of Echo signs (some recent meta-analysis have been just publicised on this issue).

No data regarding PASP was collected in order to calculate PESI-Echo;

We have also uploaded an updated version of the manuscript. 

Best regards

Reviewer 2 Report

The authors provide a timely report on PE among hospitalizations during the Covid-19 pandemic, and elaborate the possible relationship between the PE and pandemic. The work is important, though some revisions will help to improve the impact of it.

  1. The authors attempt to compare the PE rate prior to and after the pandemic, however, the local population is missing. Can the authors provide an estimate on the local population prior to and after the pandemic?
  2. In Fig 3,  Survival analysis shows the probability of survival pre-pandemic and the pandemic. However, can the authors compare these results with the literature? How is the survival rate compared with peers?
  3. Following my last question, the authors should provide error bars for the survival analysis.

Author Response

Response to reviewer no. 2

Thank you for your review. We have updated the manuscript and also provided answers to your revisions, we hope this information suits your concerns.

“The authors provide a timely report on PE among hospitalizations during the Covid-19 pandemic, and elaborate the possible relationship between the PE and pandemic. The work is important, though some revisions will help to improve the impact of it.

  1. The authors attempt to compare the PE rate prior to and after the pandemic, however, the local population is missing. Can the authors provide an estimate on the local population prior to and after the pandemic?

Our hospital is an Emergency Hospital from Bucharest, the capital of Romania, the largest city in the country. We do not have official national data regarding the changes in our local population during the pandemic, but we do have data about the number of patients admitted to our cardiology department before and during the pandemic:

Response – rows 119-127: “Between January 2019 and December 2019, 4595 patients were admitted to the cardiology unit, while between January 2020 and December 2020, only 2404 patients were hospitalized in the cardiology department (representing 52.3% of the total admissions in 2019). Among those, 107 (2.3%) patients from 2019 and 127 (5.3%) patients from 2020 had PE, respectively (p<0.0001), reflecting a higher incidence of PE among hospitalizations in the cardiology unit in the year when the COVID-19 pandemic was declared (Figure 1).  The final study cohort thus included 234 consecutive patients with PE, divided in a “pre-pandemic” (107 patients) and a “pandemic” group (127 patients). Baseline demographic and clinical characteristics are summarized in Table 1.

  1. In Fig 3,  Survival analysis shows the probability of survival pre-pandemic and the pandemic. However, can the authors compare these results with the literature? How is the survival rate compared with peers?

Response – rows 460- 478, added discussions: Watchmaker et al. published a retrospective review of PE patients from six emergency hospitals from New York City in which they compared non-pandemic and pandemic PE admissions from March 1st – April 1st 2019 and March 1st – April 1st 2020 and showed a 10.9% increase of PE presentations in the pandemic era and no enrichment in the traditional risk factors [12].

  1. Tilliridou et al. published a retrospective audit in July 2021, which assessed the frequency and severity of pulmonary embolism in 2020 compared to 2019: patients with both PE and COVID-19 had an increased 30-day mortality and worse outcomes compared to those without COVID-19 (28% versus 6%) [23]. Moreover, delays due to COVID-19 protocols all over the world were reported to forestall the treatment for acute pathologies [24].

In a retrospective multi-sourced nationwide cohort study of adults admitted to hospital with thromboembolic events (TE) and deaths from TE in England between 1st February 2018 and 31st July 2020, Aktaa et. al. found an increased incidence of TE, particularly PE, despite only a small proportion of the cohort tested positive for COVID-19 and also an increased mortality. This may highlight the adverse social effects of the pandemic and also the delay in seeking medical care which may lead to worse outcomes of the patients with thromboembolic events [25].

  1. Following my last question, the authors should provide error bars for the survival analysis.

Response – row 171-178: There were 4 in-hospital deaths in the pre-pandemic group (4%) and 14 in-hospital deaths in the pandemic group (11%), 2 of the latter occurring in patients who tested positive for SARS-CoV-2 on admission. None of the recorded in-hospital deaths occurred in patients with previous history of COVID-19. In Kaplan-Meier analysis, patients in the pandemic group had worse in-hospital survival than patients in the pre-pandemic group (log-rank test, p=0.04) (Figure 4).

Round 2

Reviewer 1 Report

No furhter comments

This manuscript is a resubmission of an earlier submission. The following is a list of the peer review reports and author responses from that submission.